# Unveiling the Potential of *Bacillus safensis* Y246 for Enhanced Suppression of *Rhizoctonia solani*

**DOI:** 10.3390/jof9111085

**Published:** 2023-11-06

**Authors:** Xing-Cheng Zhu, Shu-Gang Xu, Yu-Ru Wang, Meng-Ting Zou, Mohammed Amin Uddin Mridha, Khadija Javed, Yong Wang

**Affiliations:** 1Department of Plant Pathology, Agriculture College, Guizhou University, Guiyang 550025, China; zmyxtdcz@163.com (X.-C.Z.); m18216835457@163.com (S.-G.X.); 18508513887@163.com (Y.-R.W.); zmtmtmtmt@163.com (M.-T.Z.); 2Faculty of Graduate Studies, Daffodil International University, Birulia, Savar, Dhaka 1216, Bangladesh; mridha52@gmail.com; 3Julius Kühn-Institut (JKI) for Biological Control, 64287 Darmstadt, Germany

**Keywords:** *Bacillus safensis*, antifungal activity, *Rhizoctonia solani*, volatile organic compounds

## Abstract

*Rhizoctonia solani* is a significant pathogen affecting various crops, including tobacco. In this study, a bacterial strain, namely Y246, was isolated from the soil of healthy plants and exhibited high antifungal activity. Based on morphological identification and DNA sequencing, this bacterial strain was identified as *Bacillus safensis.* The aim of this investigation was to explore the antifungal potential of strain Y246, to test the antifungal stability of Y246 by adjusting different cultivation conditions, and to utilize gas chromatography–mass spectrometry (GC-MS) to predict the volatile compounds related to antifungal activity in Y246. In vitro assays demonstrated that strain Y246 exhibited a high fungal inhibition rate of 76.3%. The fermentation broth and suspension of strain Y246 inhibited the mycelial growth of *R. solani* by 66.59% and 63.75%, respectively. Interestingly, treatment with volatile compounds derived from the fermentation broth of strain Y246 resulted in abnormal mycelial growth of *R. solani*. Scanning electron microscopy analysis revealed bent and deformed mycelium structures with a rough surface. Furthermore, the stability of antifungal activity of the fermentation broth of strain Y246 was assessed. Changes in temperature, pH value, and UV irradiation time had minimal impact on the antifungal activity, indicating the stability of the antifungal activity of strain Y246. A GC-MS analysis of the volatile organic compounds (VOCs) produced by strain Y246 identified a total of 34 compounds with inhibitory effects against different fungi. Notably, the strain demonstrated broad-spectrum activity, exhibiting varying degrees of inhibition against seven pathogens (*Alternaria alternata*, *Phomopsis.* sp., *Gloeosporium musarum*, *Dwiroopa punicae*, *Colletotrichum karstii*, *Botryosphaeria auasmontanum*, and *Botrytis cinerea*). In our extensive experiments, strain Y246 not only exhibited strong inhibition against *R. solani* but also demonstrated remarkable inhibitory effects on *A. alternata*-induced tobacco brown spot and kiwifruit black spot, with impressive inhibition rates of 62.96% and 46.23%, respectively. Overall, these findings highlight the significant antifungal activity of *B. safensis* Y246 against *R. solani*. In addition, Y246 has an excellent antifungal stability, with an inhibition rate > 30% under different treatments (temperature, pH, UV). The results showed that the VOCs of strain Y246 had a strong inhibitory effect on the colony growth of *R. solani*, and the volatile substances produced by strain Y246 had an inhibitory effect on *R. solani* at rate of 70.19%. Based on these results, we can conclude that Y246 inhibits the normal growth of *R. solani*. These findings can provide valuable insights for developing sustainable agricultural strategies.

## 1. Introduction

Tobacco, a solanaceous plant primarily cultivated in regions such as South America, South Asia, and China, holds significant economic value as a cash crop, particularly in China. However, the production of tobacco has always been hindered by the crop’s susceptibility to various diseases, which has led to substantial losses in yield and quality. Notably, fungal diseases have posed significant challenges to tobacco cultivation. For instance, *Alternaria alternata* has been identified as the causal agent of tobacco brown spot disease [1], *Colletotrichum fructicola* as the cause of anthracnose disease [2], and *Rhizoctonia solani* as the culprit behind tobacco target disease [3]. Recently, tobacco target spot disease caused by *R. solani* was observed to proliferate in varying degrees in several regions of Guizhou province [4]. Research has revealed that *R. solani* has a broad host range and demonstrates strong saprophytic activity [5]. These findings underscore the urgency of addressing target spot disease in tobacco production. In addition, this particular pathogen, known to cause significant harm to various crops such as Gramineae, Solanaceae, Leguminosae, and Brassicaceae, has been associated with several destructive plant diseases. For instance, it has been identified as the causal agent behind rice sheath blight [6], bottom rot of lettuce [7], stem and fruit rot in tomato [8], and tobacco target spot disease [9]. 

Tobacco target spot is a disease first identified by Costa in Brazil in 1948 [10]. It has been increasingly detected in various regions since the 1980s [11]. Notably, the occurrence of tobacco target spot disease has been observed in Costa Rica, the Carolinas, Italy, and several other locations [12]. In 1985, Shew referred to the disease as *Rhizoctonia* leaf spot [13], but it was renamed “tobacco target spot” in 1990 [14]. Over the past decade, tobacco target spot disease has been discovered and reported in China, particularly in Guangxi, Heilongjiang Province, Yunnan, Anhui, Sichuan, Hunan, and other places [15]. Consequently, it has become crucial to identify effective treatment methods for this disease. Chemical control has traditionally been the primary approach for managing plant diseases, with chemically synthesized fungicides being a significant tool in reducing disease severity. However, the widespread and excessive use of chemical agents has resulted in the emergence of fungus-resistant pathogens, environmental pollution, and concerns about food safety [16]. To address these issues, the development of safe and environmentally friendly bio-fungicides has become a reliable solution. Biological control, which harnesses beneficial microorganisms to combat harmful ones, has garnered considerable attention due to its safe and environmentally friendly nature [17]. This approach is considered to be a green, safe, and efficient means of disease prevention and control, holding great potential for sustainable and environmentally friendly disease management in plants [18,19]. In the natural environment, the crop rhizosphere has been a valuable source for the isolation of biocontrol agents. Microorganisms such as *Pseudomonas* [20], *Trichoderma* [21], *Streptomyces* [22], and *Bacillus* [23] have been successfully identified and proven effective in preventing and controlling crop diseases. By utilizing these beneficial microorganisms, researchers have demonstrated the efficacy of biological control in disease management.

*Bacillus* species, a diverse group of Gram-positive bacteria, are nature’s silent guardians, wielding their formidable powers to protect against the relentless threat of *R. solani*, a plant pathogen. For instance, Kang [24] discovered that *B. amyloliquefaciens subsp. plantarum* GR53 significantly inhibits *Rhizoctonia* infection in cabbage. Similarly, Peng [25], in addition to having observed the high effectiveness of *B. subtilis* NJ-18 in inhibiting *R. solani*, also reported that combining *B. subtilis* NJ-18 with Jinggangmycin had a more pronounced inhibitory effect on rice sheath blight as compared to using either component alone. The combination of *B. subtilis* NJ-18 at a concentration of 10^8^ cfu mL^–1^ and Jinggangmycin at 50 mg L^−1^ resulted in a 35% reduction in disease severity. Furthermore, *B. subtilis*, *B. velezensis*, and *B. amyloliquefaciens* have demonstrated beneficial effects on the growth of various plants, and exhibit antagonistic properties against pathogens affecting cucumber [26], rice [27], and tobacco [28]. Overall, *Bacillus* spp. play a vital role in the prevention and control of plant diseases, offering multiple benefits to plants through their diverse mechanisms of action.

Further research has demonstrated that volatile substances produced by *Bacillus* species can also exhibit inhibitory effects on plant pathogens [29]. Numerous studies have specifically highlighted the presence of volatile organic compounds such as 2-Tridecanone and 2-Decanone [30], 2-Heptanone [31], and 2,3-Butanedione [32] in *B. subtilis* that inhibit pathogens. As research progresses, an increasing number of volatile substances with antimicrobial properties are being discovered. Gao et al. found that a volatile organic compound in the fermentation broth of *B. subtilis* CF-3 cultured for 24 h could inhibit the mycelial growth of *Botrytis cinerea*, *Colletotrichum gloeosporioides*, *Penicillium expansum*, *Monilinia fructicola*, and *A. alternata* [33]. Asari et al. reported that *B. amyloliquefaciens* UCMB5113 produced higher amounts of pyrazines in TSA and LBA media, which led to the improved inhibition or stimulation of *B. cinerea* by *B. amyloliquefaciens* [34]. Morita et al. found that *B. pumilus* TM-R exhibited strong antifungal activity against five foodborne fungi, namely *A. alternata*, *Cladosporium cladosporioides*, *Curvularia lunata*, *Fusarium oxysporum*, and *Penicillium italicum*, with a particularly high inhibition rate (93%) against *Penicillium italicum*, during the storage of agricultural products [35]. Although there are limited reports on the use of *Bacillus* spp. for the control of tobacco target spot disease, it is crucial to explore new biological control strains and strategies that demonstrate robust and consistent efficacy against this disease. Therefore, the objectives of this study are as follows: (1) to screen and identify strains that exhibit significant antagonistic effects against *R. solani*; (2) to assess the inhibitory effects of bacterial suspensions and fermentation broths derived from strain Y246 on *R. solani*; (3) to investigate the broad-spectrum antifungal activity of strain Y246; (4) to determine the activity of volatile substances produced by the strain Y246; (5) to examine changes in the cell morphology of strain Y246 when exposed to pathogens using scanning electron microscopy (SEM), and (6) to predict whether the volatile compounds produced by the selected strain have antifungal effects through gas chromatography–mass spectrometry (GC-MS) analysis.

## 2. Materials and Methods

### 2.1. Preparation of the Pathogen and Antifungal Strain

The pathogen *R. solani* Kühn (BG3) was obtained from the Department of Plant Pathology, Guizhou University, China. To prepare the pathogen, BG3 was inoculated onto potato dextrose agar (PDA) plates and incubated at 28 °C for 4–5 days. The resulting colonies were then preserved in a 15% glycerol solution at −4 °C until further use. An antagonistic strain, namely Y246, was isolated from the soil found between the roots of healthy plants in areas with infected tobacco in Qingzhen, ZunYi, Guizhou Province, China. The isolation process involved the mixing 10 g of soil sample with 90 mL of sterile water, which was then homogenized at 37 °C and 200 rpm for 20 min. Subsequently, the samples were diluted to four different concentrations using the plate dilution method. Each dilution was evenly spread (100 μL) on a nutrient agar (NA) medium, and the plates were incubated at a temperature of 30 °C ± 1 °C for 12 h to allow for the growth of single colonies. The single colony was preserved in a 30% glycerol solution and stored at −80 °C.

### 2.2. Screening of the Antifungal Strain

The isolated strains were tested in vitro for their antagonistic effect against *R. solani* as per the method developed by Yan et al. [36]. In brief, the pathogen (BG3) was isolated, purified, and then cultured at 28 °C for 4–5 days. The antagonistic strain was inoculated into an LB agar medium by lineation and cultured at 30 °C for 12 h until a single colony was obtained. After that, the single colony was inoculated into an LB liquid culture medium and kept for 12 h in orbital incubator at 37 °C and 200 rpm. The strain was centrifuged at 10,000 rpm for 1 min, after which the supernatant was removed, and 100 μL of sterile water was added and mixed until the concentration was 1 × 10^8^ cfu/mL^−1^. Then, a disk 5 mm in diameter was taken from the margin of each actively growing *R. solani* colony and brought to the center of the PDA plate. Subsequently, 1 μL of suspension was inoculated at four symmetrical points located 25 mm from the pathogen. All plates were incubated at 28 °C for 4–5 d. The growth inhibition rate of mycelia was calculated using the following formula: i = (a_1_ − a_2_)/a_1_ × 100, where i is the growth inhibition rate of mycelia, a_1_ is the hyphae area of the untreated pathogen, and a_2_ is the hyphae area of the treated pathogen. The screened antagonistic strains were then preserved in glycerol. All experiments were performed in triplicate.

### 2.3. Identification of Antagonistic Strain

The antagonistic strain was isolated and identified based on its morphology and DNA sequence. The strains were cultured on LB medium plates at 28 °C for 12–16 h. Gram staining was performed on the colonies and observed under a light microscope to determine their shape. Genomic DNA from strain Y246 was extracted using the EZ-10 column bacterial genomic DNA extraction kit (Shanghai Biotechnology Technology Services Co., Ltd., Shanghai, China). PCR amplification of the 16S rDNA and gyrB region was carried out using universal primers, i.e., 16SrDNA primer (27F 5′-AGAGTTTGATCCTGGCTCAG-3′ and 1492R 5′-GGTTACCTTGTTACGACTT-3′) and gyrB primer (gyrB-F 5′-TTATCTACGACCTTAGACG-3′ and gyrB-R 5′-TAAATTGAAGTCTTCTCCG-3′), and the resulting PCR products were analyzed via 1.2% gel electrophoresis. The PCR products were sequenced at the Shanghai Biotechnology Company. The obtained sequences were verified and assembled using the Contig Express software version 3.0 (Copyright 1999–2000 InforMax, Inc.). To determine the strain relationship, the sequences were compared with the GenBank nucleic acid database using the National Center for Biotechnology Information BLAST software (NCBI), (Basic Local Alignment Search Tool (nih.gov). In addition, interspecific sequences were downloaded from existing literature for comparison. The IQ-TREE website was used to construct a phylogenetic tree using the maximum likelihood (ML) method. The combined analysis of colony morphology and molecular biology identification confirmed the classification of strain Y246.

### 2.4. Activity Spectrum of the Strain 

To determine the broad-spectrum antifungal activity of strain Y246, seven pathogens were tested in this study. The pathogenic fungi *Alternaria alternata*, *Phomopsis.* sp., *Gloeosporium musarum*, *Dwiroopa punicae*, *Colletotrichum karstii*, *Botryosphaeria auasmontanum*, and *Botrytis cinerea* were provided by the Department of Plant Pathology at Guizhou University in Guiyang, China. The antifungal spectrum activity of strain Y246 was determined following the methodology outlined in Section 2.2, with minor modifications. To assess the antifungal spectrum of the strain, a 5 mm disk of the selected pathogen was placed at the center of a PDA plate. Then, a 1 μL suspension of the strain was inoculated at four symmetrical points located 25 mm away from the pathogen. Each experiment was performed in triplicate to ensure reliable results. The plates were incubated at a temperature of 28 °C for 5 d. After the incubation period, the inhibition rate was measured using the procedure described in Section 2.2. 

### 2.5. Preparation of Y246 Fermentation Broth and Bacterial Suspension

Strain Y246 was cultured in an LB medium and placed in a shaker at 37 °C and 220 rpm for 12 h to obtain the antagonistic bacterial fermentation broth (BF). Afterward, the fermentation broth was centrifuged at 10,000× *g* for 10 min to separate the supernatant and bacterial components. The supernatant was discarded, and the bacterial portion was suspended in sterile ultrapure water to obtain the bacterial suspension (BS).

### 2.6. Evaluation of BS and BF on R. solani Growth Inhibition

The method to obtain BF and BS described above was employed to assess the potential inhibitory effects of the strain on the target pathogen that causes spotted disease. LB liquid was used as a control. The culture plates were then transferred to an incubator and kept for cultivation at 28 °C for 4–5 d. Upon the growth of the mycelium of the control group in the culture dish, the colony diameter for each treatment group was measured. The strain inhibition rate was determined using the following formula: i = (a_1_ − a_2_)/a_1_ × 100, where i represents the growth inhibition rate of the mycelia, a_1_ denotes the hyphae area of the untreated pathogen, and a_2_ represents the hyphae area of the treated pathogen. All experiments were performed in triplicate to ensure accuracy and reliability.

### 2.7. Aseptic Filtrate Stability

In order to investigate the antifungal changes in strain Y246 under different treatments and verify its antifungal stability, we selected different temperatures, ultraviolet rays, and pH to treat the strain. Firstly, the sterile fermentation broth of Y246 was subjected to water bath treatment at various temperatures (30, 50, 70, 80, and 90 °C) for a duration of 30 min. A blank control group was included without being subjected to the water bath treatment. In addition, the initial pH of the sterile fermentation broth was adjusted using hydrochloric acid and sodium hydroxide, and the final pH levels were set at 1, 3, 5, 7, 9, and 11. The broth was left for 20 min, after which the pH of the treatment group was neutralized. The blank control group consisted of a sterile fermentation broth without any pH adjustment. For the antifungal strain, the sterile fermentation broth was exposed to ultraviolet (UV) radiation at different durations (10, 30, 70, 90, and 110 min). A separate blank control group was prepared with a fermentation broth that did not undergo the UV radiation treatment. After the above experiments, *R. solani* was introduced into the center of each plate. The methodology described in Section 2.2 was followed, and each experiment was repeated three times. The diameter of *R. solani* was measured to calculate the inhibition rate.

### 2.8. Effects of Y246 Volatiles on Mycelial Growth

To test whether the Y246 of volatile substances could inhibit the growth of the pathogen *R. solani*, the antifungal activity of volatile organic compounds (VOCs) in the mycelial growth of *R. solani* was assessed using the double plate method. In a controlled environment and on an ultra-clean table, *R. solani* was extracted using a 5 mm hole punch and placed at the center of a potato dextrose agar (PDA) plate, designated as plate A. The fermentation broth was prepared for strain Y246 using the method described in Section 2.5. Subsequently, a cultured bacteria fermentation liquid was added to the LB agar medium at a ratio of 1:100 when the temperature dropped to 45 °C. This mixture was then poured onto another plate (plate B) and inverted. Plate A (PDA plate with pathogen) was positioned at the bottom, and plate B (LB plate mixed with fermentation liquid) was placed on top. The plates were sealed and incubated at 28 °C for 4–5 days. Plate B with sterile water was used as a control. The experiment was replicated three times. The diameter of the *R. solani* colony was measured, and the inhibition of mycelium growth was calculated using the following formula: i = (a_1_ − a_2_)/a_1_ × 100, where i represents the growth inhibition rate of mycelia, a_1_ is the hyphae area of the untreated pathogen, a_2_ is the hyphae area of the treated pathogen; the diameter of the control was multiplied by 100.

### 2.9. Scanning Electron Microscope

To investigate the morphological changes in *R. solani* mycelia induced by volatile substances, two sets of samples were prepared: untreated mycelia and mycelia of strain Y246 treated for 12 h. The mycelia were washed with 0.1 M PBS (pH 7.4) 3 times at 15 min each, and 1% O_S_O_4_ was used for post-fixing at 24 °C for 1–2 h. After fixation, the mycelia were dehydrated using a series of ethanol solutions (30, 50, 70, 80, 90, and 100%) at 15 min each. The dehydrated mycelia were then subjected to critical point drying (Critical Point Dryer, K850, Quorum, Lewes, UK) with CO_2_ and coated with gold (Ion Sputtering Apparatus, MC1000, Hitachi, Tokyo, Japan). The samples were finally examined using a scanning electron microscope (SU8100, Hitachi, Tokyo, Japan).

### 2.10. Identification of Y246 VOCs with GC-MS

The volatile organic compounds (VOCs) of strain Y246 were identified using gas chromatography–mass spectrometry (GC-MS) technology. Purified Y246 was inoculated into a 15 mL centrifuge tube containing 5 mL of LB liquid medium, and the tube was placed in a rotating shaker at 37 °C and 200 rpm for 12 h. Subsequently, 50 μL of the bacterial fermentation broth was uniformly applied to a 20 mL brown headspace bottle containing 5 mL of tilted LB medium. To ensure consistent experimental conditions, the headspace bottle was then incubated at 28 °C for 4–5 d, with a blank LB medium serving as the control. This experimental setup was repeated three times to ensure reproducibility. For the separation and identification of VOCs from the Y246 strain, an Agilent 7890B GC (Santa Clara, CA, USA) coupled with a 5977B MS was utilized. The online detection of strains was performed under the following conditions: incubation temperature of 50 °C, shaking time of 15 min, incubation time of 30 min, desorption time of 5 min, shaking speed of 250 rpm, DB-Wax column (30 m × 0.25 mm × 0.25 μm), column flow rate of 1 mL min^−1^, electron energy of 70 eV, mass range of *m*/*z* 20–400, and scan mode set to Full Scan.

### 2.11. Statistical Analyses

The data were subjected to statistical analysis using Excel 2010 and SPSS version 25 (SPSS Inc., Chicago, IL, USA). To determine the significant differences at a *p*-value of less than 0.05, a one-way ANOVA was performed, followed by Duncan’s multiple range test. Charts were created using Origin 2022.

## 3. Result and Discussion

### 3.1. Screening and Identification of Antagonistic Strain

We isolated strains from healthy soil found in areas with infected tobacco in Qingzhen, Guizhou, China. Through preliminary and subsequent screenings, we identified seven strains that exhibited inhibitory activity against BG3 (Figure 1). Among these, strain Y246 displayed strong inhibitory effects against *R. solani*, suppressing the pathogen’s mycelial growth by 76.30% (Figure 1 and Table 1). Further experiments were conducted with Y246 with the highest inhibition. First, to determine its species, morphological analysis and Gram staining of strain Y246 were conducted after culturing in an LB medium for 24 h. The bacterial colonies of strain Y246 exhibited a viscous state, with a milky white color, irregular edges, and a distinct odor (Figure 2a). Gram staining and microscopic observation confirmed that the biocontrol bacteria were Gram-positive, short rod-shaped cells (Figure 2b,c). To further classify strain Y246, we performed sequencing and phylogenetic analysis on its 16SrDNA (Figure 3A) and gyrB (Figure 3B). The results of the 16S rDNA and gyrB sequencing for strain Y246 revealed it to be 100% *Bacillus safensis* (login number: OR492439) (Figure 2). Based on the colony characteristics, 16S rDNA and gyrB sequencing, and homology analysis, we preliminarily identified the biocontrol strain Y246 as *B. safensis*.

### 3.2. Antifungal Spectrum of Strain Y426

To determine whether strain Y246 exhibits broad-spectrum antifungal activity, we used seven pathogens in the tests. The results showed that Y246 was capable of inhibiting the mycelial growth of all tested pathogens. Figure 4 and Table 2 provide detailed information on the mycelial growth inhibition rates of Y246. In Table 2, it can be seen that Y246 exhibited mycelial growth inhibition rates that exceeded 36.11%. One noteworthy finding was the significant inhibitory effect (62.96%) of the strain on the tobacco brown spot pathogen (Figure 4(a1,a2)). However, strain Y246 demonstrated relatively lower inhibitory rates against apple fruit rot disease, with a value of 36.11% (Figure 4(d1,d2)). Overall, the results suggest that strain Y246 exhibits broad-spectrum antifungal activity against various fungal pathogens, with particularly strong inhibitory effects on the tobacco brown spot pathogen (Figure 4 and Table 2). 

### 3.3. Effect of Strain Y246 on the Mycelial Growth of R. solani

By measuring the effects of the bacterial suspension and fermentation broth of strain Y246, we found that the fermentation broth and bacterial suspension exhibited inhibitory effects on the mycelium of *R. solani* (Figure 5). Notably, the Y246 fermentation broth showed the highest inhibition rate at 66.59% (Figure 5a and Table 3). The bacterial suspension of Y246 also demonstrated a relatively high inhibition rate of 63.75% (Figure 5b and Table 3). Based on the above results, strain Y246 was considered for the following experiments.

### 3.4. Aseptic Filtrate Stability

The antifungal effects of the Y246 fermentation broth were evaluated at different temperatures, pH, and UV. In Figure 6, it can be seen that the antifungal activity of the fermentation filtrate of strain Y246 remains at 58–74% at pH 7–9, with the strongest antifungal activity reaching 58% at pH 7. It can also maintain over 40% activity in strong acidic and alkaline environments at pH values of 3 and 11 (Figure 6b). This indicates that the stability of its acid and alkali resistance is high, but its activity is the highest in neutral environments. It is thus advisable to maintain a neutral environment in practical applications. After the ultraviolet (UV) treatment, the inhibitory effect of the Y246 fermentation filtrate on *R. solani* decreased slightly, but the difference was not significant (*p* < 0.01), and the antifungal rate was above 54%, indicating that UV had no significant effect on the antifungal activity of the Y246 fermentation filtrate (Figure 6c). When we tested whether the fermentation broth of strain Y246 could withstand high temperature changes, the results showed that the fermentation broth of strain Y246 had varying degrees of inhibitory effects on the hyphae of tobacco target spot pathogens under different temperature treatments. As the treatment temperature increased, the inhibition rate of the strain only showed slight changes. When the processing temperature reached 90 °C, the inhibitory effect of the fermentation broth of strain Y246 on the pathogen of tobacco target spot disease reached up to 40% At temperatures as high as 110 °C, the inhibition rate of strain Y246 also reached over 37% (Figure 6a). This indicates that the fermentation broth of strain Y246 has strong thermal stability and an ideal processing temperature.

### 3.5. The Effect of VOCs on R. solani

In order to further investigate strain Y246, the plate buckle method was used to determine whether the volatile substances of Y246 could inhibit *R. solani*. The results showed that the volatile organic compounds of strain Y246 had a strong inhibitory effect on the colony growth of *R. solani* (Figure 7b), while the control colony grew normally (Figure 7a). In Figure 7, it can be seen that the volatiles of Y246 have an inhibitory effect on the growth of the tested pathogen hyphae. According to the inhibition rates in Table 4, the volatile substances produced by strain Y246 have an inhibitory effect on *R. solani* at rate of 70.19%. Based on these results, we can conclude that Y246 inhibits the normal growth of *R. solani*.

### 3.6. SEM (Scanning Electron Microscopy)

To further validate the antifungal effect of Y246 on *R. solani*, we observed *R. solani* mycelium treated with Y246 under a 40× microscope. We found that the mycelium grew sparsely, was twisted and deformed, and had an uneven thickness (Figure 8b). Afterwards, we conducted an SEM analysis, as depicted in Figure 8c,d. The findings revealed the untreated mycelium to have a smooth and structurally intact appearance (Figure 8c). In stark contrast, the mycelium of *R. solani* treated with Y246 for 12 h displayed notable morphological damage (Figure 8d). Specifically, the mycelium appeared bent and deformed, with rough and protruding surfaces. These outcomes are in line with the morphological changes observed through the optical microscope (Figure 8b). Based on the results of the plate experiment and the observed hyphal malformation under the microscope and SEM, strain Y246 significantly inhibits the normal growth of *R. solani*.

### 3.7. Analysis of VOCs by GC-MS

A non-targeted metabolomics analysis was conducted using GC-MS to analyze volatile organic compounds (VOCs) of strain Y246. The results, presented in Table 5 and Figure 9, revealed the detection of a total of 243 volatile organic compounds through the utilization of MetaboAnalyst 5.0. Among the 243 compounds, 34 were identified as having beneficial properties. These included 2,3-Butanedione [37], 2-Heptanone [31], and 2,5-dimethyl, Pyrazine, methyl- [38], etc. The results of this study justify future research to determine the antifungal activity of some of the identified VOCs in order to evaluate their potential bioactivity against antibiotic-resistant bacteria.

## 4. Discussion

At present, chemical control is the main method utilized for the treatment of common fungal diseases during the growth cycle of tobacco, with high application frequency and the difficult removal of residues contributing to its significant impact on the environment and tobacco quality [39]. Consequently, the use of physical control has increased; however, the effectiveness of large-scale physical control is not satisfactory. Therefore, developing efficient, broad-spectrum, and stable microbial biofungicides has important practical significance [40]. At present, there are few reports on the prevention and control of *R. solani* with the use of *Bacillus* in China. Strain Y246 screened in this study has been proven to have high antagonistic activity against the main diseases of tobacco and has great development and application value. In this study, the antifungal broad spectrum of strain Y246 was tested by selecting different pathogenic fungi. We found that among seven different pathogenic fungi, the fermentation broth of strain Y246 showed significant inhibition against *A. alternata* (62.96%); thus, strain Y246 can not only be applied to the control of *R. solani*, but is also effective against plant diseases caused by *A. alternata*, such as tobacco brown spot and kiwifruit black spot diseases. The fermentation broth of strain Y246 showed a more stable inhibition rate against the target pathogens after high temperature, different pH and UV irradiation times, which provided a foundation for the next step, which is to make strain Y246 into a fungicide. In addition, the extraction and identification of volatile organic compounds (VOCs) using gas chromatography-mass spectrometry (GC-MS) showed that strain Y246 produces various VOCs, including 2,3-Butanedione, 2,4-ditert butylphenol, 3-methyl-1-butanol, 1-nonanol, 2-decanol, 2-decanone, among others, which have been reported to have different effects on plant growth, nematicidal activity, and antifungal properties. For example, Zhao et al. studied the inhibitory effect of volatile organic compounds on the defenses of litchi fruit against *C. gloeosporioides*, and the results of their study showed that 2,4-ditert-butylphenol could inhibit the activity of pathogenic enzymes secreted by *C. gloeosporioides* as well as activate antioxidant enzymes and disease-resistant enzymes to enhance the fruit’s resistance to *C. gloeosporioides* and inhibit its growth [41]. Furthermore, Luo et al. demonstrated the in vivo and in vitro biocontrol effects of VOCs released by *B. velezensis* L1 on *A. iridiaustralis*, a pathogenic fungus responsible for wolfberry fruit rot. The study found that 2.3-butanedione had the strongest antifungal effects, which completely inhibited *A. iridiaustralis* in wolfberry fruit at a concentration of 60 μL/L [42]. Gamboa-Becerra studied that lateral root formation of *Arabidopsis thaliana* was regulated by 4-methyl-2-pentanone, 3-methyl-1-butanol, 1-nonanol, and ethyl isovalerate, suggesting a participation via JA signaling [43]. In addition, Cheng et al. found that furfural acetone, 2-decanol, and 4-acetylbenzoic acid have contact nematicidal activity against *M. incognita* [44]. Among the volatile substances produced by strain Y246, 2-decanone is considered to have a very significant antifungal effect. Yuan et al. reported that *B. amyloliquefaciens* NJN-6 also produced 2-decanone volatile organic compounds and in their study of *Fusarium oxysporum*, the results showed that 2-decanone had 100% inhibitory effects against the pathogen [45].

Currently, chemical control methods dominate the management of common fungal diseases during the tobacco growth cycle [46]. However, these methods require frequent application and result in the accumulation of difficult-to-remove residues, thereby adversely affecting the environment and tobacco quality [29]. Physical control methods are also employed but have proven to be unsatisfactorily effective on a large scale. Therefore, the development of efficient, broad-spectrum, and stable microbial bio fungicides holds crucial practical significance. In China, there have been limited reports on the utilization of *Bacillus* for the prevention and control of tobacco target spot disease. Strain Y246 identified in this study demonstrated high inhibition activity against *R. solani* and has considerable potential for further development and application. The next step can be the study of volatile substances detected from the strain, which would further clarify whether the use of volatile substances of the strain is effective against *R. solani*, and could provide data supporting the prevention and control of this disease.

## 5. Conclusions

The main focus of this study was on the screening of the biocontrol strain taken from the rhizosphere soil and its inhibition of the pathogen *R. solani*, which causes tobacco target spot disease. In this work, we obtained a bacterial strain that exhibited high inhibition of *R. solani.* Through morphological observation and molecular biology identification, it was preliminarily determined that strain Y246 belongs to *B. safensis*. We conducted experiments to determine whether the suspension, fermentation broth, broad-spectrum antifungal activity, and volatile substances produced by strain Y246 could inhibit *R. solani.* The results indicate that the fermentation broth and suspension of Y246 exhibit varying degrees of antagonistic activity against *R. solani.* With respect to the antifungal spectrum of Y246, varying degrees of antagonistic activity against seven pathogens were exhibited. Strain Y246 had a higher inhibition rate of 62.96% against *A. alternata*. Moreover, the fermentation broth of strain Y246 demonstrated remarkable acid–base stability and resistance to UV radiation. It also exhibited high temperature tolerance, demonstrating its potential as a bacterium capable of withstanding elevated temperatures. In addition, through GC-MS extraction, 243 volatile organic compounds were detected in strain Y246, some of which have been reported to have antifungal effects. Based on the above results, we believe that strain Y246 can effectively inhibit *R. solani*, and that it is a strain worthy of further study.

## Figures and Tables

**Figure 1 jof-09-01085-f001:**
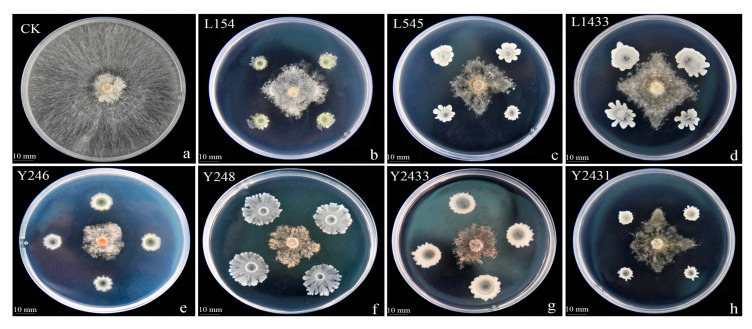
Preliminary screening of biocontrol bacterial strains against *Rhizoctonia solani*: (**a**) Control *Rhizoctonia solani*; (**b**) strain L154; (**c**) strain L545; (**d**) strain L1433; (**e**) strain Y246; (**f**) strain Y248; (**g**) strain Y2433; (**h**) strain Y2431.

**Figure 2 jof-09-01085-f002:**
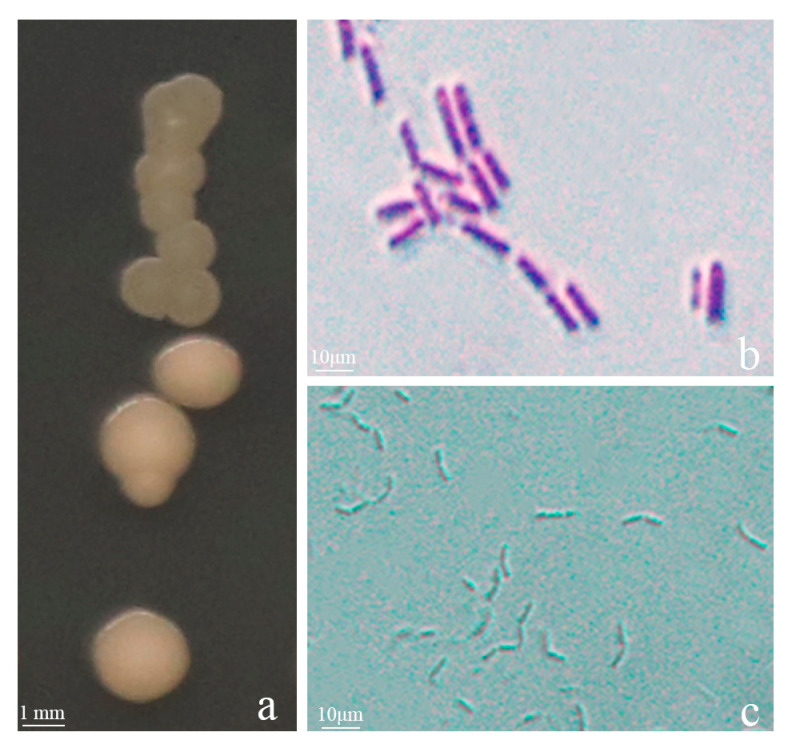
Identification of strain Y246. (**a**) Morphology of strain Y246. (**b**) Strain Y246 is Gram positive. (**c**) Strain Y246 under 40× microscope.

**Figure 3 jof-09-01085-f003:**
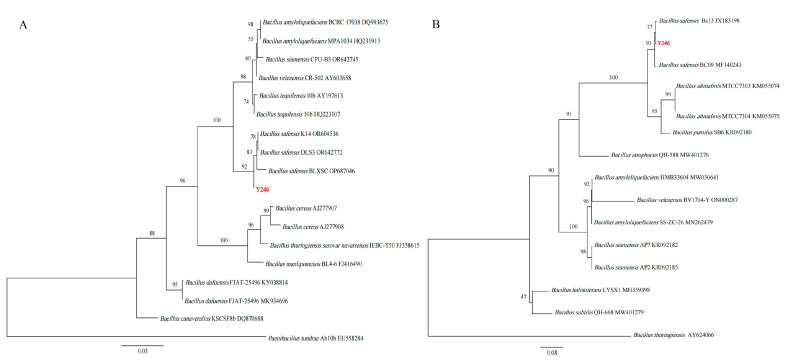
Phylogenetic trees of Y246: (**A**) based on 16s; (**B**) based on gyrB. Note: The antagonistic strain Y246 is highlighted in red.

**Figure 4 jof-09-01085-f004:**
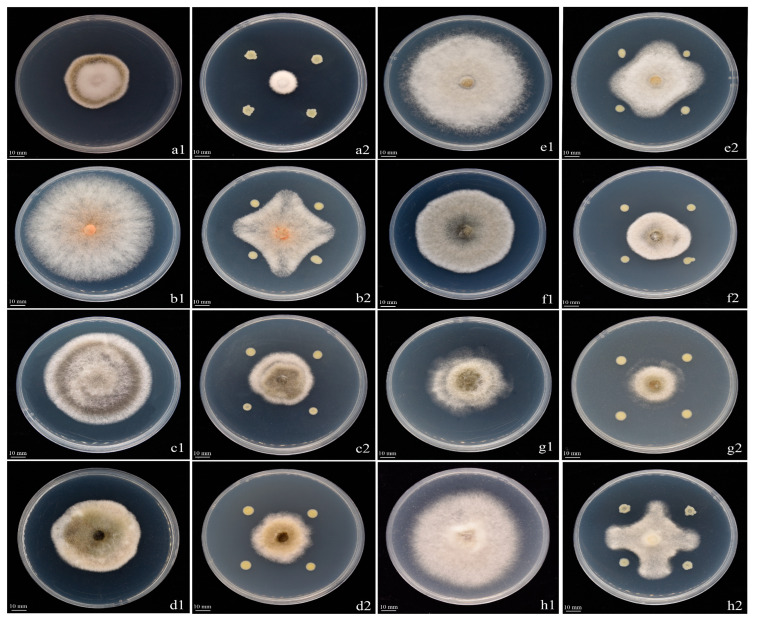
Strain Y246 inhibits seven different pathogens: (**a1**) Control (**a2**) *Alternaria alternata*, (**b1**) Control (**b2**) *Gloeosporium musarum*, (**c1**) Control (**c2**) *Dwiroopa punicae*, (**d1**) Control (**d2**) *Botryosphaeria auasmontanum*, (**e1**) Control (**e2**) *Phomopsis*. sp., (**f1**) Control (**f2**) *Alternaria alternata*, (**g1**) Control (**g2**) *Colletotrichum karstii*, (**h1**) Control (**h2**) *Botrytis cinerea*.

**Figure 5 jof-09-01085-f005:**
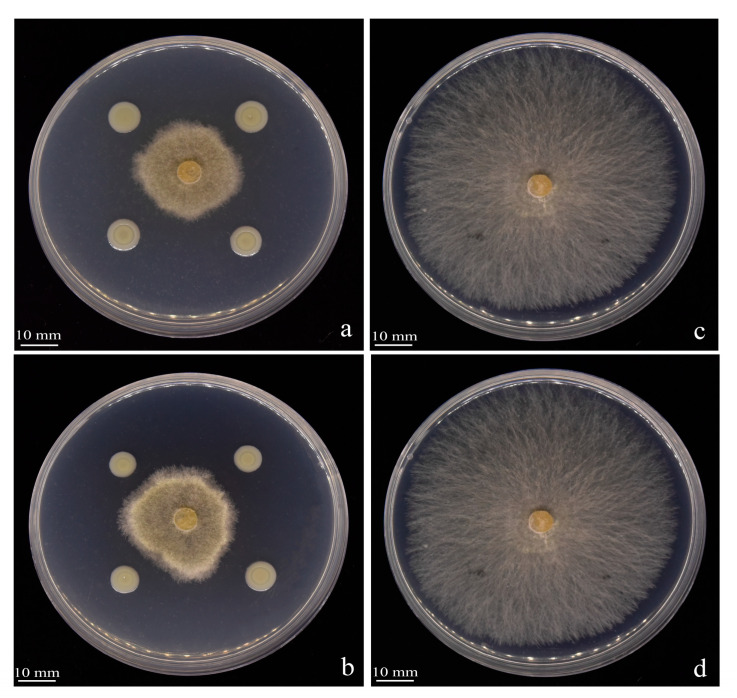
Effects of strain Y246 fermentation broth and bacterial suspension on *Rhizoctonia solani.* (**a**) Fermentation broth of Y246; (**b**) bacterial suspension of Y246; (**c**,**d**) control *Rhizoctonia solani*.

**Figure 6 jof-09-01085-f006:**
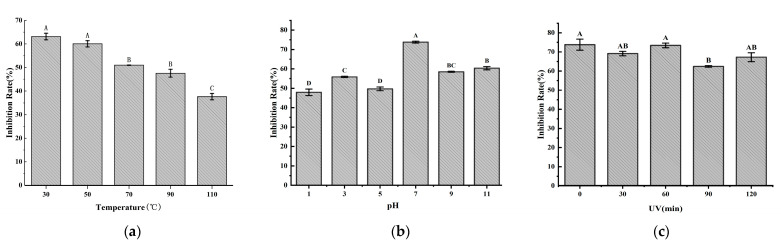
Stability of Y246 fermentation broth treated with different temperatures (**a**), pH values (**b**), and UV irradiation time (**c**). The numerical value is represented as the average of three replicates ± SE. The same letters above the bars mean that there is no significant difference between treatments (*p* < 0.01).

**Figure 7 jof-09-01085-f007:**
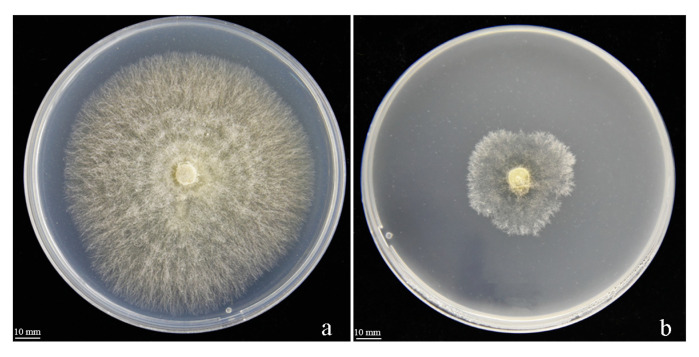
The effect of VOCs of strain Y246. (**a**) Control *Rhizoctonia solani*; (**b**) strain Y246.

**Figure 8 jof-09-01085-f008:**
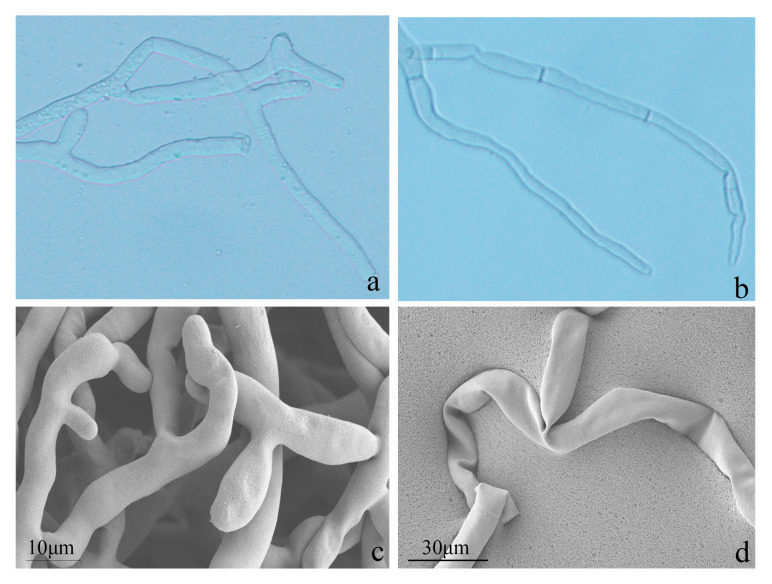
(**a**,**b**) Optical microscopy (40×) and (**c**,**d**) scanning electron micrograph analysis of *Rhizoctonia solani* mycelia ultrastructure morphology: (**a**,**c**) control; (**b**,**d**) strain Y246.

**Figure 9 jof-09-01085-f009:**
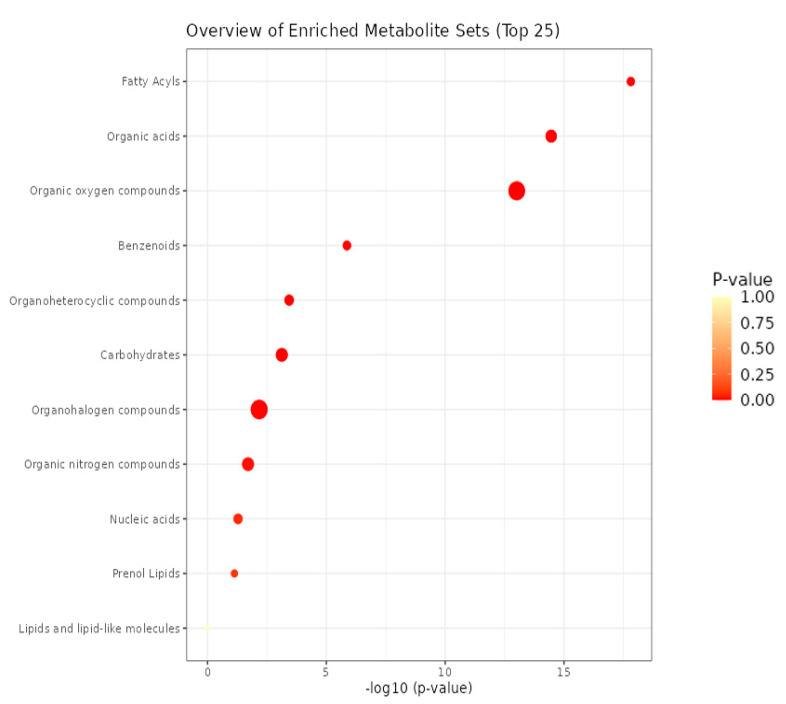
MetaboAnalyst-based enrichment analysis of VOCs of *B. safensis* strain Y246.

**Table 1 jof-09-01085-t001:** Preliminary screening results of biocontrol bacterial strains against *R. solani.*.

Strain Code	Colony Diameter (mm)	Inhibition Rate (%)
Control	90.00	/
L154	32.33 ± 1.16	64.07 ± 1.29
L545	35.33 ± 3.83	60.74 ± 4.26
L1433	41.33 ± 3.56	54.07 ± 3.96
Y246	21.33 ± 1.48	76.30 ± 1.64
Y248	25.15 ± 0.59	72.06 ± 0.66
Y2433	24.65 ± 2.15	72.61 ± 2.39
Y2431	33.83 ± 1.20	62.41 ± 1.33

Numerical values are expressed as mean ± standard error (SE) of triplicates.

**Table 2 jof-09-01085-t002:** Antifungal spectrum of strain Y246.

Disease	Pathogen	Inhibition Rate of Y246 (%)
Tobacco brown spot	*Alternaria alternata*	62.96 ± 0.02
Kiwifruit soft rot	*Phomopsis.* sp.	41.54 ± 0.54
Banana anthracnose	*Gloeosporium musarum*	40.04 ± 0.54
Kiwifruit black spot	*Alternaria alternata*	46.23 ± 0.75
Pomegranate leaf spot	*Dwiroopa punicae*	37.92 ± 0.14
Anthracnose of passion fruit	*Colletotrichum karstii*	44.32 ± 0.81
Fruit rot of apple	*Botryosphaeria auasmontanum*	36.11 ± 0.53
Tomato gray mold	*Botrytis cinerea*	40.08 ± 0.60

Numerical values are expressed as mean ± standard error (SE) of triplicates.

**Table 3 jof-09-01085-t003:** Inhibition rate of Y246 fermentation broth and bacterial suspension on *R. solani*.

Treatment	Colony Diameter (mm)	Inhibition Rate (%)
Fermentation broth	25.78 ± 0.42	66.59 ± 0.54
Bacterial suspension	27.97 ± 2.61	63.75 ± 3.39

Numerical values are expressed as mean ± standard error (SE) of triplicates.

**Table 4 jof-09-01085-t004:** Effects of VOCs on strain of Y246.

Treatment	Colony Diameter (mm)	Inhibition Rate (%)
CK	72.63 ± 0.21	/
Y246	21.65 ± 0.69	70.19 ± 0.94

Numerical values are expressed as mean ± standard error (SE) of triplicates.

**Table 5 jof-09-01085-t005:** Reported active substances in the volatile organic compounds (VOCs) of *B. safensis* strain Y246.

Compounds ID	Name	Molecular Formula	Retention time (min)
Com_11	Methyl Isobutyl Ketone	C_7_H_16_O	5.5
Com_15	2-Butanone	C_4_H_8_O	3.29
Com_24	3-Hydroxypropionic acid	C_3_H_6_O_3_	35.77
Com_27	Butanoic acid, 3-methyl-	C_5_H_10_O_2_	24.40
Com_30	1-Nonanol	C_9_H_22_O	14.26
Com_51	Dodecane	C_12_H_26_	11.42
Com_62	2-Nonanone	C_9_H_18_O	18.80
Com_68	2,3-Butanedione	C_4_H_6_O_2_	4.73
Com_94	Benzaldehyde	C_7_H_6_O	20.64
Com_97	2-Decanol	C_10_H_22_O	20.33
Com_101	2-Heptanone	C_7_H_14_O	11.01
Com_104	1-Butanol	C_4_H_10_O	10.33
Com_110	2-Ethyl-1-hexanol	C_8_H_18_O	7.91
Com_120	Pyrazine, methyl-	C_5_H_6_N_2_	13.71
Com_133	Dibutyl phthalate	C_16_H_22_O_4_	42.80
Com_143	1-Butanol, 3-methyl-	C_7_H_14_O_2_	12.20
Com_147	2,3-Butanediol	C_4_H_10_O_2_	19.02
Com_148	Butyrolactone	C_24_H_24_O_8_	23.20
Com_160	Ethyl Acetate	C_4_H_8_O_2_	3.13
Com_167	Propanoic acid, 2-methyl-	C_4_H_8_O_2_	21.98
Com_172	Butanoic acid, 2-methyl-	C_5_HO_2_	24.38
Com_175	2-Octanol	C_8_H_18_O	18.29
Com_179	Furfural	C_5_H_4_O_2_	19.15
Com_181	Acetoin	C_4_H_8_O_2_	14.29
Com_183	Disulfide, dimethyl	C_2_H_6_S_2_	7.39
Com_195	Tridecane	C_13_H_28_	14.70
Com_198	2-Decanone	C_10_H_20_O	19.10
Com_201	Hexanal	C_6_H_12_O	7.76
Com_210	Propanoic acid	CH_3_CH_2_COOH	21.23
Com_215	Ethanol	C_2_H_6_O	3.93
Com_219	Acetic acid	C_2_H_4_O_2_	18.98
Com_237	2-Octanone	C_8_H_16_O	14.23
Com_242	Pyrazine, 2,5-dimethyl-	C_6_H_8_N_2_	15.41
Com_243	Heptadecane	C_17_H_36_	25.13

## Data Availability

The required data set is already available in manuscript file; other data sets generated during the study are available upon request from corresponding author.

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
