# Peer review of "Unveiling the Potential of Bacillus safensis Y246 for Enhanced Suppression of Rhizoctonia solani"

_jof, 2023, doi:10.3390/jof9111085_

Round 1

Reviewer 1 Report

Comments and Suggestions for Authors

All suggestions are listed in the file with manuscript. 

Comments on the Quality of English Language

Reviewer 2 Report

Comments and Suggestions for Authors

Microorganisms as biological antifungal agents play important roles in developing effective strategies for sustainable agriculture to replace synthetic chemical fungicides. This study is interesting and may contribute to practical applications.

Comments:

- Revise the title of the manuscript as "Unveiling the Potential of Bacillus safensis Y246 for Enhanced Suppression of Rhizoctonia Solani" -> "Biocontrol Bacterium" is redundant.

- Line 14: "the strain Y246" -> should be "a bacterial strain, namely Y246"

- Line 19: "high inhibition" -> should be clearly mentioned: "high fungal inhibition"

- Line 24: "the stability of the fermentation broth" -> difficult to understand, you mean "the stability of antifungal activity of the fermentation broth"? 

- Line 26: why "antibacterial" here, should be "antifungal or antimicrobial"

- Line 26: "strain's robust stability" -> what do you mean? stability of anfungal activity?, should be described clearly.

-  Lines 28-29: "different fungi", "six pathogens" -> what are they? the names of these fungal pathogens should be listed here.

- Lines 31-32: These sentences should be rewritten, do not repeat the same fungus, just combine. The fungal pathogen Alternaria alternata causes  brown spot and black spot in two diferrent plant hosts.

- Line 33: "the antifungal stability of Y246 is good" -> what do you mean "good", 60%, 70% or 80%?

- Lines 33-43: "The predicted VOCs were found to have biocontrol agent" -> too general, not informative, not understandable.

- Keywords:  I don't think "Gas chromatography-mass spectrometry, Scanning electron microscopy" are considered as keywords. Add some other keywords: volatile organic compounds,  Rhizoctonia solani, ...

The authors must check errors, mistakes, typos throughout the manuscript to correct them or revise the whole sentences or paragraphs. There are many expression errors. Many sentences are not meaningful or understandable. Some sentences are even not correct, for instance:

Line 80: "Bacillus spp. is a type of Gram-positive bacteria". Remember that "spp." = "species plural". It should not be a singular noun. Line 91: similar.

Line 88: "Bacillus belles" ???? I have never heard about this species of Bacillus. For this situation, I think this species should be "Bacillus velezensis".

Line 115: why "antibacterial activity"???? -> should be "antifungal activity". Please check the whole manuscript and correct them. It is definitely wrong.

Line 116: "by the said strain"??? -> should be "the tested strain, the reported strain, strain Y246, ...)

Line 141: why do you use "a concentrator" for microbial cultivation with a speed? This should be a orbital incubator or shaker.

And too too many mistakes like I mention above. 

- Identification of strain Y246: PCR and sequencing of 16S rRNA is not enough to identify Y246 as Bacillus safensis. In the Bacillus genus, five closely related species, including Bacillus pumilus, B. safensis, B. stratosphericus, B. altitudinis, and B. aerophilus. These species are difficult to distinguish due to their 99.5% similarity in their 16S rRNA gene sequence.

-> For accurate identification, I suggest the authors to sequence one of two genes for DNA gyrase (gyrase A or gyrase B gene). Only with this sequence result, you can conclude correctly strain Y246 as Bacillus safensis or not.

Comments on the Quality of English Language

Too many sentences are difficult to understand what the authors mean, and many mistakes. Extensive editing of English language is required.

Round 2

Reviewer 1 Report

Comments and Suggestions for Authors

All suggestions from reviewers are accepted.

Reviewer 2 Report

Comments and Suggestions for Authors

The revised manuscript is now suitable for publication in the journal. 

I have no further comments.